

# The CH-IRP data set: a decade of fortnightly data of $\delta^2$H and $\delta^{18}$O in streamflow and precipitation in Switzerland

Maria Staudinger[1], Stefan Seeger[2], Barbara Herbstritt[2], Michael Stoelzle[3], Jan Seibert[1], Kerstin Stahl[3], Markus Weiler[2]

[1]Hydrology and Climate Unit, Department of Geography, University of Zurich, 8057, Zurich, Switzerland
[2]Chair of Hydrology, University of Freiburg, 79098, Freiburg, Germany
[3]Chair of Environmental Hydrological Systems, University of Freiburg, 79098, Freiburg, Germany

*Correspondence to*: Maria Staudinger (maria.staudinger@geo.uzh.ch)

**Abstract.**

The stable isotopes of oxygen and hydrogen, $^2$H and $^{18}$O, provide information on water flow pathways and hydrologic catchment functioning. Here a data set of time series data on precipitation and streamflow isotope composition in Swiss medium-sized catchments, CH-IRP, is presented that is unique in terms of its long-term multi-catchment coverage along an alpine to pre-alpine gradient. The data set comprises fortnightly time series of both $\delta^2$H and $\delta^{18}$O as well as Deuterium excess from streamflow for 23 sites in Switzerland, together with summary statistics of the sampling at each station. Furthermore, time series of $\delta^{18}$O and $\delta^2$H in precipitation are provided for each catchment derived from interpolated datasets from the NISOT, GNIP and ANIP networks. For each station we compiled relevant metadata describing both the sampling conditions as well as catchment characteristics and climate infomation. Lab standards and errors are provided, and potentially problematic measurements are indicated to help the user decide on the applicability for individual study purposes. For the future, it is planned that the measurements will be continued at 14 stations as a long-term isotopic measurement network and the CH-IRP data set will, thus,

be continuously be extended. The data set can be downloaded from data repository zenodo

https://doi.org/10.5281/zenodo.3659679 (Staudinger et al., 2020).

## 1  Introduction

There are significant differences in the isotopic contents of seawater, freshwater (Gilfillan, 1934), rain

and snow. The isotopic composition in precipitation further depends on meteorological influences such

as air temperature, rainfall amount and intensity, and location parameters such as altitude, latitude and

distance from the coast (Dansgaard, 1953, 1964; Epstein, 1956; Friedman, 1953).

When tracing the path of water through a hydrological system such as a catchment, the composition of

the stable water isotopes $\delta^{18}O$ and $\delta^{2}H$ of precipitation changes by the time it reaches the stream. The

isotopic signal in the precipitation is changed along the water flow pathways through a catchment

resulting in a temporal delay and a dampened amplitude of the signal in the streamflow. This signal

change can be modelled using water transit time distributions or other approaches to consider water ages,

and can, hence, help improve the understanding of hydrological functioning of catchments. Event-based

isotope sampling has long been the basis for hydrograph separation in hydrological research and allows

quantifying pre-event and event water contributions to soil water, streamflow, or groundwater

(Christophersen et al., 1990; Klaus and McDonnell, 2013; Sprenger et al., 2019). More extended time

series of the isotopic composition of catchment discharge, i.e., streamflow, allow the estimation of water

transit times and storage od catchments (McGuire and McDonnell, 2006). Besides their value to develop,

calibrate and validate a wide variety of catchment hydrological models, these data sets also have a

demonstrated value for catchment intercomparisons in, for instance, Sweden (Lyon et al., 2010), Oregon

(McGuire et al., 2005) and different Northern regions (Tetzlaff et al., 2009). McGuire et al. (2005) used

isotopic data for a three-year period to quantify mean transit times for seven, partly nested, catchments in

the Cascades, Oregon, US, and found good relations to topographic indices such as the catchment average

of L/G, where L is the distance from the stream and G is the gradient to the stream. In a similar study in

Northern Sweden based on 15 snow-dominated subcatchments of the Krycklan catchment, Lyon et al.

(2010) found wetlands to be a controlling factor for catchment transit times. Tetzlaff et al. (2009)

compiled a data set of 55 catchments in different regions in the northern temperate zone. Their analysis

showed that topography is an essential control in catchments with a pronounced topography, whereas the

topographic influences are smaller in regions with a flatter topography. In the latter, hydrological soil

characteristics become relatively more important.

Here, we present a long-term data set consisting of $\delta^2$H and $\delta^{18}$O values for streamflow and precipitation

for 23 catchments in Switzerland and discuss the applicability of the data. The collection of this data set

started in 2010 as part of the DROUGHT-CH project (Seneviratne et al., 2013) and is still continuing.

## 2    CH-IRP data set

## 2.1    $\delta^2$H and $\delta^{18}$O for streamflow

### 2.1.1    Data sampling

The data were collected in 23 catchments with near-natural streamflow in Switzerland. The catchments

were selected based on two different criteria and two different temporal sampling resolutions were chosen.

The majority of catchments were selected with the focus on studying and comparing low-flow behaviour.

Therefore, we selected catchments without major water abstractions or transfers, where the gauging

stations     provided     precise     streamflow     measurements     also     during     low     flows     (see

https://opendata.swiss/de/dataset/niedrigwasserstatistik-nqstat and Marti and Kan (2003)). These

catchments vary in size, mean elevation, topographic characteristics as well as underlying geology (Table

1, **Figure 1**). For these catchments the sampling was done fortnightly.

Five catchments belong to the Alptal long-term hydrological research catchments (Alp, Biber, Erlenbach,

Lümpenenbach, and Vogelbach). Here, grab samples were collected fortnightly until December 2014 and

have been collected weekly starting from January 2015.

Additional isotope data were collected within other research projects in the Alptal catchments (**Figure 1**,

inset zoom) during events with higher temporal resolution over a short period or in snapshot campaigns

with higher spatial coverage. Fischer et al. (2016) performed isotopic hydrograph separations for several

events in five small headwater catchments in the Alptal and found that the event-water fraction depended

much more on the event size than on catchment characteristics. These findings contributed to the

emerging conceptual understanding of runoff generation in the Alptal (van Meerveld et al., 2018). A

general observation for isotopes in the Alptal is the large spatial variation which was found for both,

rainfall (Fischer et al., 2017) and groundwater (Kiewiet et al., in press). Rücker et al. (2019) measured

the isotopic composition of snowpack outflow to study runoff generation during rain-on-snow events.

Furthermore, in the Alptal a field lab was installed that provides isotope data at high temporal resolution

at the outlet of the Erlenbach catchment (von Freyberg et al., 2018). The data stemming from theses

studies is not part of the CH-IRP data set but could be useful for specific research questions.

All samples were taken as grab samples using 100 mL High Density PolyEthylene (HD-PE) bottles. The

sampling personnel was instructed to flush the bottle with stream water before taking the sample and to

ensure to tightly close the bottle to minimize exchange with the atmosphere and thus to avoid fractionation

of the samples.

### 2.1.2   Lab analysis

All liquid water samples were analyzed at the Chair of Hydrology at the University of Freiburg, Germany. The laboratory regularly participates successfully in IAEA Water Isotope Inter-Comparisons (WICO) (Wassenaar et al., 2018). The samples were analyzed using CRDS laser spectrometers (either Picarro

L2120-*i* or L2130-*i*, Picarro Inc., Santa Clara, CA, USA) in 'high precision mode'. Samples were filtered via syringe filters (0.45 µm) prior to analysis if they were muddy. Of each sample, 1 mL was filled into autosampler vials. According to the manufacturer handbook, six injections per vial were analyzed with the isotope analyzer and raw data of the first three injections were discarded to keep memory effects from one sample to the next at a minimum. Mean and standard deviation (SD) of the last three injections were

calculated. In case there was still a memory effect and the SD was larger than 0.08‰ (in the case of $\delta^{18}O$) or larger than 0.30‰ (in the case of $\delta^2H$), the fourth injection was also discarded and only the last two injections were averaged.

Calibration of the raw data was then conducted using three in-house standards with distinct isotopic compositions, -14,86‰, -9.47‰, and 0.30‰ for $\delta^{18}O$, -107.96‰, -66.07‰, and 1.53‰ for $\delta^2H$,

referenced to the international VSMOW-SLAP scale (Craig, 1961). The standards were analyzed in triplicates each and averaged. The light and the heavy standards – embracing the samples – were used for a 2-point calibration, the third standard was used for validation. Long-term post-calibration accuracy of the validation standard was ± 0.05‰ for $\delta^{18}O$ and ± 0.35‰ for $\delta^2H$. Typically, the nine standards were evenly distributed between 40 samples and thereby additionally used to check for instrument drift. Besides



the calibration of the measurements to an international system, a comparison to the standard is useful

because it allows an implicit consideration of all corrections for instrumental effects and interferences.

Furthermore, most of the influences are cancelled out since they affect both the sample and the reference

standard equally (Gourcy et al., 2005).

All isotope data are expressed in δ-notation calculated following Eq. (1) (after Gonfiantini (1981)):

$$\delta = \left( \frac{R_{sample}}{R_{VSMOW}} - 1 \right) \cdot 1000\text{‰} \tag{1}$$

where *VSMOW* is the Vienna Standard Mean Ocean Water and *R* is the isotope ratio ($^{18}O/^{16}O$ or $^{2}H/^{1}H$).

During times when there was enough lab capacity, double measurements were performed and the

arithmetic average was calculated (for ~50% of the samples).

### 2.1.3    Summary statistics

The sampling periods for streamflow was about eight years for 14 catchments, about three years for seven

catchments and for the remaining two catchments five years and 1.5 years, respectively. There is an

overlap of three years with data for almost all catchments (~90%), and for the 14 stations that are still

being sampled more than  eight years of overlapping data are available. Since the stations were sampled

fortnightly the number of samples was between 26 and 224 per station, while for the Alptal catchments

the samples were taken weekly from 2015 on and these catchments have a total number of 318 samples

each.

We performed a statistical outlier analysis based on z-scores (from a visual inspection of the data sets

using qq-plots we assume the data are normally distributed). There were in total 47 outliers according to

the z-score in either $\delta^2H$ and $\delta^{18}O$ or both with an absolute value larger than three, indicating that the

value deviates more than three standard deviations from the mean.

Isotopic compositions can strongly deviate during high flow conditions compared to mean or baseflow conditions because of larger proportion of event and more recent precipitation, as was found for instance in the high-resolution dataset of the Plynlimon catchment, Wales (Knapp et al., 2019). The time series of $\delta^2H$ in **Figure 2** and of $\delta^{18}O$ **Figure 3** give an overview of the seasonal changes for each catchment in the

isotope composition. The relationship of $\delta^2H$ and $\delta^{18}O$ does not deviate for any of the catchments from the global meteoric water line (GMWL, $\delta^2H = 8.0\ \delta^{18}O + 10‰$) (**Figure 4**).

### 2.1.4    Streamflow conditions during sampling

In addition to the values of $\delta^2H$ and $\delta^{18}O$ and deuterium excess we provide an index of sampling conditions regarding streamflow (sampling Q index). This index was calculated from the sum the

streamflow volume on the day of sampling and the previous two days divided by the sum of the long-term mean streamflow over the same days of the year. An index larger than one indicates wetter conditions from the long-term mean, an index smaller than one indicates drier conditions than the long-term mean. This information can also be assembled to analyze the frequency of samples taken under certain streamflow conditions as given by the index (**Figure 7**).

Furthermore, we calculated the flow percentile on the flow duration curve during the sampling to get an idea if the samples were taken during baseflow, average or high flow conditions. A percentile with the value 0.95, for instance, would be exceeded 95 % of the time during the year and indicates very low streamflow conditions. The percentiles were calculated empirically based on the last 20 years of the available record of streamflow (1999 - 2018), note that this period contained some pronounced low flow





periods (2003, 2011, 2015 and 2018). These computations demonstrated that the percentiles for the

sampling times in each catchment span the full range of percentiles, which indicates that samples were

taken during baseflow, mean flow and high flow conditions. Most samples were taken during mean flow

conditions.

Comparing the mean catchment elevation shows that the isotope values follow the elevation gradient

(Figure 5 and Figure **6**) as expected (Dansgaard, 1964). The Riale di Calneggia catchment shows an

exception to this general gradient. This catchment in the Canton Ticino in the Southern Alps receives

precipitation from the Mediterranean Sea and thus shows a less depleted isotopic signal. The elevational

gradient that is visible in the long-term mean values of the time series is also visible in specific seasons

(JFM, AMJ, JAS, OND) and is very similar for $\delta^2$H and $\delta^{18}$O.

## 155    2.2    $\delta^2$H and $\delta^{18}$O for precipitation

For isotopes in precipitation, data from the National Network for the Observation of Isotopes in the Water

Cycle (ISOT, see Schotterer, 2010) was used to predict precipitation isotopes for the selected catchments.

For the catchments close to the Swiss border also data from the Austrian network (ANIP) as well as the

global network (GNIP) were used to allow for a better interpolation. Average elevation gradients for each

month were calculated from a representative gradient based on three NISOT stations (**Figure 1**). Monthly

$\delta^{18}$O values corrected to sea level elevation were computed for every measurement site of the original

network by multiplying the average elevation gradient for the month with the elevation of the respective

NISOT station and adding it to the isotopic signal measured at this site. The corrected average monthly

$\delta^{18}$O values were interpolated in space by Kriging (Delhomme, 1978, implemented for R in the gstat-

package by Pebesma, 2004) resulting in continuous monthly maps (example for $\delta^{18}$O in **Figure 8**). In



order to obtain an estimate of precipitation isotopic signature for any desired location, the deviation of the closest available measurement site from its long term average value of the respective month was combined with the interpolated map of corrected average values for the respective month. A more detailed description of the interpolation method can be found in Seeger and Weiler (2014).

Due to limited availability of precipitation isotope data time series with an appropriate length, the data were only validated qualitatively (Seeger and Weiler, 2014), by comparing the predicted isotope values to limited time series of sites situated in North-Eastern and Central Switzerland. The comparison between the interpolated data and validation data suggested good agreement. However, explorative simulations by Seeger and Weiler (2014) also showed a bias of up to 2‰ for $\delta^{18}$O between the interpolated precipitation
values and the measured discharge values. This suggests that the interpolated values are well suited to predict the amplitudes of the temporal variations of the precipitation isotopes, while the steep topography of the Swiss Alps might lead to regional inhomogeneities that are not fully captured by the data underlying the interpolation procedure.

### 2.3 Data file format

$\delta^2$H and $\delta^{18}$O in streamflow are provided as one ASCII.txt file for each station. Additionally to these time series each of the files contains the Deuterium excess, the streamflow conditions preceding the sampling (Q sampling index and streamflow percentiles) as well as the z-scores indicating if a sample might be a statistical outlier, assuming the data are normally distributed. All files contain further information for each sample whether double measurement was performed in the lab comments indicating for instance
special sampling conditions (ice etc.) or storage-related issues that could alter the isotopic composition due to fractionation (e.g., sample bottle not closed tightly).

For each data file for streamflow data there is a corresponding ASCII.txt file for catchment precipitation. These contain the interpolated $\delta^2$H and $\delta^{18}$O in precipitation for the catchment as well as the source data that were used to derive the interpolated values.

## 3    Data application and outlook

This data set with isotope data from precipitation and streamflow allows the estimation of mean transit times. With these, catchment water storage (mobile storage) can be estimated and may be related to the sensitivity to droughts (Staudinger et al. 2017). From this data set also young water fractions can be calculated and for instance, using the data presented here, Von Freyberg et al. (2018) assessed how sensitive the young water fraction is to both hydro-climatic forcing and catchment properties. Using the $\delta^{18}$O values in precipitation and streamflow for 12 catchments of the presented data set Allen et al. (2019) assessed whether summer or winter precipitation is overrepresented in streamflow, relative to its proportion of total precipitation. Also parts of this data set (composition of isotopes in precipitation) were used to re-investigate the relationship between transit times and catchment topography (Seeger and Weiler, 2014).

In 2019 we were still collecting data for 14 sites and it is planned to continue these observations. A long-term sampling will allow for more robust estimations of storages and young water, and hence for a more robust reassessment of the mentioned studies. The growing dataset will also provide opportunities for a closer look at catchment transit times and storages. The data set will, for instance, allow us to compare different conditions such as dry or wet years or the effects of extreme events.

Additional data that may be useful for potential applications are time series of streamflow time series and shapefiles of the catchments, which are both provided by the Swiss FOEN, meteorological time series, which are provided by MeteoSwiss, and a digital elevation model of Switzerland, which is provided by Swisstopo.

## 210  4  Data availability

The data set can be downloaded from the data repository zenodo https://doi.org/10.5281/zenodo.3659679 (Staudinger et al., 2020).

## Author contribution

MS, JS, KS, MW designed the sampling net, BH and MS did the lab analysis and maintained the data bank. SS interpolated the isotopes in precipitation. MS wrote the first draft of the manuscript. All authors contributed to the discussion and revised the submitted manuscript.

## Competing interests

The authors declare that they have no conflict of interest.

## Acknowledgements

We thank all the persons that went out to take samples, many of them starting in 2010 continuing until today. We want to explicitly mention Mr. M. Altermatt, Tiefbauamt Basel-Land, H. Aschwanden and R.
Gamma, Tiefbauamt Kanton Uri and Kari Steiner, WSL. Financial support for this long-term sampling

campaign came from the Swiss National Foundation, NFP61 "Drought-CH" (2010 - 2016), the Swiss

Federal Office for the Environment (FOEN) (Project "Auswirkungen der Klimaveränderung auf das

Grundwasser und Niedrigwasser, November 2013-December 2016, contract 13.0055.PJ/M372-1682) and

the University of Zurich. The basic data for the interpolation of isotopic compositions in precipitation

came from NISOT provided by FOEN, the National Groundwater Monitoring, NAQUA, from GNIP

provided by IAEA as well as ANIP from the Environment Agency Austria. The streamflow data were

provided by FOEN and Office of Waste, Water, Energy and Air, WWEA, Kanton Zurich.

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

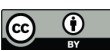



**Table 1 Catchment characteristics:** ID is the identification number that is used throughout the paper. FOEN ID is the identification number that is used by the Swiss Federal Office of the Environment (FOEN), coordinates are given using the official Swiss reference system CH1903+, aquifer productivity is given as relative catchment area with low, varying and high productivity (Bitterli, 2004).

| ID | Catchment | FOEN ID | Coord. x | Coord. y | Size km² | Elevation mean | Elevation max | Elevation min | Slope mean [°] | Slope <3° [%] | Slope >15° [%] | Glacier [%] | Aquifer low [%] | Aquifer Varying [%] | Aquifer High [%] | Streamflow detailed | Streamflow simple |
|---|---|---|---|---|---|---|---|---|---|---|---|---|---|---|---|---|---|
| MEN | Mentue | 2369 | 545440 | 180875 | 105.00 | 683 | 946 | 436 | 5 | 35 | 3 | 0 | 100 | 0 | 0 | pluvial jurassien | pluvial |
| ROE | Roethenbach | | 627081 | 191605 | 53.00 | 1000 | 1539 | 727 | 12 | 8 | 26 | 0 | 99 | 0 | 1 | nivo-pluvial préalpin | nivo-pluvial |
| SEN | Sense | 2179 | 593350 | 193020 | 352.00 | 780 | 1044 | 524 | 9 | 12 | 14 | 0 | 85 | 2 | 13 | nivo-pluvial préalpin | nivo-pluvial |
| GUE | Guerbe | 2472 | 605890 | 181880 | 53.70 | 1038 | 2152 | 566 | 17 | 10 | 50 | 0 | 76 | 1 | 23 | pluvial supérieur | pluvial |
| ALL | Allenbach | 2232 | 608710 | 148300 | 28.80 | 1855 | 2833 | 1093 | 24 | 1 | 77 | 0 | 88 | 3 | 9 | nival alpin | snow |
| ILF | Ilfis | 2603 | 627320 | 198600 | 188.00 | 1047 | 2045 | 681 | 17 | 4 | 56 | 0 | 92 | 0 | 8 | nivo-pluvial préalpin | nivo-pluvial |
| LAN | Langeten | 2343 | 629560 | 219135 | 59.90 | 765 | 1123 | 566 | 8 | 11 | 8 | 0 | 77 | 13 | 10 | pluvial inférieur | pluvial |
| ERL | Erlenbach | | 680200 | 155350 | 0.60 | 1359 | 1650 | 1117 | 15 | 0 | 43 | 0 | 82 | 18 | 0 | nivo-pluvial préalpin | nivo-pluvial |
| RIA | Riale di Calneggia | 2356 | 684970 | 135960 | 24.00 | 1982 | 2866 | 645 | 32 | 1 | 93 | 0 | 96 | 4 | 0 | nival méridional | snow |
| VOG | Vogelbach | | 697020 | 214700 | 1.55 | 1335 | 1540 | 1038 | 20 | 0 | 81 | 0 | 100 | 0 | 0 | nivo-pluvial préalpin | nivo-pluvial |
| LUE | Luempenen | | 695060 | 212520 | 0.93 | 1336 | 1508 | 1092 | 12 | 0 | 24 | 0 | 99 | 1 | 0 | nivo-pluvial préalpin | nivo-pluvial |
| BIB | Biber | 2604 | 697240 | 223280 | 31.90 | 1008 | 1515 | 602 | 11 | 18 | 32 | 0 | 94 | 0 | 6 | pluvial supérieur | pluvial |
| ALP | Alp | 2609 | 698640 | 223020 | 46.40 | 1008 | 1515 | 602 | 11 | 18 | 32 | 0 | 73 | 17 | 10 | nivo-pluvial préalpin | nivo-pluvial |
| AAM | Aa | | 702145 | 242884 | 55.60 | 523 | 859 | 402 | 5 | 44 | 2 | 0 | 85 | 0 | 15 | pluvial inférieur | pluvial |
| MUR | Murg | 2126 | 714105 | 261720 | 78.90 | 654 | 1113 | 456 | 9 | 23 | 19 | 0 | 86 | 1 | 13 | pluvial inférieur | pluvial |
| RIE | Rietholzbach | 2414 | 718840 | 248440 | 3.30 | 794 | 938 | 671 | 14 | 1 | 42 | 0 | 100 | 0 | 0 | pluvial inférieur | pluvial |
| AAC | Aach am Bodensee | 2312 | 744410 | 268400 | 48.50 | 476 | 609 | 391 | 3 | 67 | 0 | 0 | 87 | 0 | 13 | pluvial inférieur | pluvial |
| DIS | Dischmabach | 2327 | 786220 | 183370 | 43.30 | 2294 | 3180 | 1545 | 26 | 1 | 84 | 1 | 91 | 9 | 0 | b-glacio-nival | snow |
| OVA | Ova da Cluozza | 2319 | 804930 | 174830 | 26.90 | 2361 | 3115 | 1468 | 31 | 0 | 92 | 0 | 8 | 15 | 77 | nivo-glaciaire | snow |
| EMM | Emme | 2409 | 627910 | 191180 | 124.00 | 1283 | 2161 | 562 | 18 | 2 | 55 | 0 | 90 | 0 | 10 | nivo-pluvial préalpin | nivo-pluvial |
| ERG | Ergolz | 2202 | 622270 | 259750 | 261.00 | 591 | 1181 | 296 | 12 | 10 | 34 | 0 | 42 | 28 | 30 | pluvial jurassien | pluvial |
| SCH | Schaechen | 2491 | 692480 | 191810 | 109.00 | 1722 | 3221 | 436 | 28 | 1 | 85 | 2 | 74 | 1 | 25 | nivo-glaciaire | snow |
| SIT | Sitter | 2112 | 749040 | 244220 | 74.20 | 1219 | 2431 | 445 | 20 | 6 | 54 | 0 | 56 | 3 | 41 | nival de transition | nivo-pluvial |




**Table 2 Inventory of the data availability per catchment.**

| ID | Catchment | Period from | to | Total sample number | Length (years) | Average sampling (1/days) | Number of comments |
|----|-----------|------|-----|------|------|------|------|
| MEN | Mentue | 05/07/2010 | 27/02/2013 | 63 | 2.7 | 15 | 0 |
| ROE | Roethenbach | 23/06/2010 | 13/11/2013 | 70 | 3.4 | 18 | 0 |
| SEN | Sense | 11/11/2011 | ongoing | 198 | 8.2 | 15 | 6 |
| GUE | Guerbe | 08/07/2010 | 18/12/2012 | 64 | 2.4 | 14 | 0 |
| ALL | Allenbach | 20/07/2010 | ongoing | 173 | 8.3 | 18 | 5 |
| ILF | Ilfis | 04/07/2010 | ongoing | 224 | 8.5 | 14 | 18 |
| LAN | Langeten | 02/07/2010 | ongoing | 197 | 8.5 | 16 | 1 |
| ERL | Erlenbach | 31/05/2010 | ongoing | 318 | 8.5 | 10 | 2 |
| RIA | Riale di Calneggia | 18/07/2010 | 20/12/2012 | 55 | 2.4 | 16 | 0 |
| VOG | Vogelbach | 21/06/2010 | ongoing | 318 | 8.5 | 10 | 1 |
| LUE | Luempenen | 31/05/2010 | ongoing | 330 | 8.2 | 9 | 1 |
| BIB | Biber | 31/05/2010 | ongoing | 318 | 8.6 | 10 | 1 |
| ALP | Alp | 31/05/2010 | ongoing | 319 | 8.6 | 10 | 7 |
| AAM | Aa | 06/09/2010 | 22/02/2016 | 95 | 5.5 | 21 | 0 |
| MUR | Murg | 07/07/2010 | ongoing | 128 | 8.5 | 24 | 0 |
| RIE | Rietholzbach | 23/07/2010 | 28/02/2013 | 68 | 2.6 | 14 | 0 |
| AAC | Aach am Bodensee | 23/06/2010 | 07/09/2012 | 26 | 1.4 | 20 | 0 |
| DIS | Dischmabach | 16/08/2010 | 16/09/2013 | 128 | 8.2 | 23 | 0 |
| OVA | Ova da Cluozza | 16/08/2010 | 16/09/2013 | 65 | 3.1 | 17 | 0 |
| EMM | Emme | 23/06/2010 | 13/11/2013 | 84 | 3.4 | 15 | 0 |
| ERG | Ergolz | 22/07/2010 | ongoing | 223 | 8.4 | 14 | 0 |
| SCH | Schaechen | 05/04/2011 | ongoing | 181 | 7.7 | 16 | 14 |
| SIT | Sitter | 02/11/2010 | ongoing | 185 | 8.2 | 16 | 0 |



Figure captions:

Figure 1 Map of the sampling locations and related catchments in Switzerland. The zoom shows the 5 Alptal catchments. Black dots indicate the source stations for the interpolation of the isotopic compositions in precipitation. The source of the underlying relief map is the Swiss Federal Office of Topography.

Figure 2 Fortnightly time series of δ2H. The colors indicate the regime type to which the catchments can 330    be assigned. Samples that have a comment in the data are plotted as symbols without border.

Figure 3 Fortnightly time series of δ18O. The colors indicate the regime type to which the catchments can be assigned. Note the different y-axis scaling for the nival catchments. Samples that are flagged in the data are are plotted as symbols without border.

Figure 4 Comparison of the relation between δ18O and δ2H to the global meteoric water line (GWML). 335    The colors indicate the regime type to which the catchments were assigned. Note the different x-axis scaling for the nival catchments.

Figure 5 Median and ranges (10th and 90th percentile) of δ2H for the full sampling period (left) and separately for the seasons (right) of the samples for each catchment against the mean catchment elevation. The colors indicate the regime type to which the catchments were assigned.

Figure 6 Median and ranges (10th and 90th percentile) of δ18O for the full sampling period (left) and separately for the seasons (right) of the samples for each catchment against the mean catchment elevation. The colors indicate the regime type to which the catchments were assigned.

Figure 7 Sampling conditions distribution expressed as sampling Q index. The dashed line shows the longterm streamflow conditions of the day of sampling. Values of the index larger than one indicate wetter 345    than the observed long-term conditions, whereas values smaller one indicate conditions drier than the long-term at this day of the year (sum of three consecutive days). For the Roethenbach catchment (ROE) simulated streamflow was used. Note that the x-axis is log-scaled to visualize the percentual difference more intuitively.

Figure 8 Monthly maps of sea level corrected δ18O in precipitation.



Figure 1

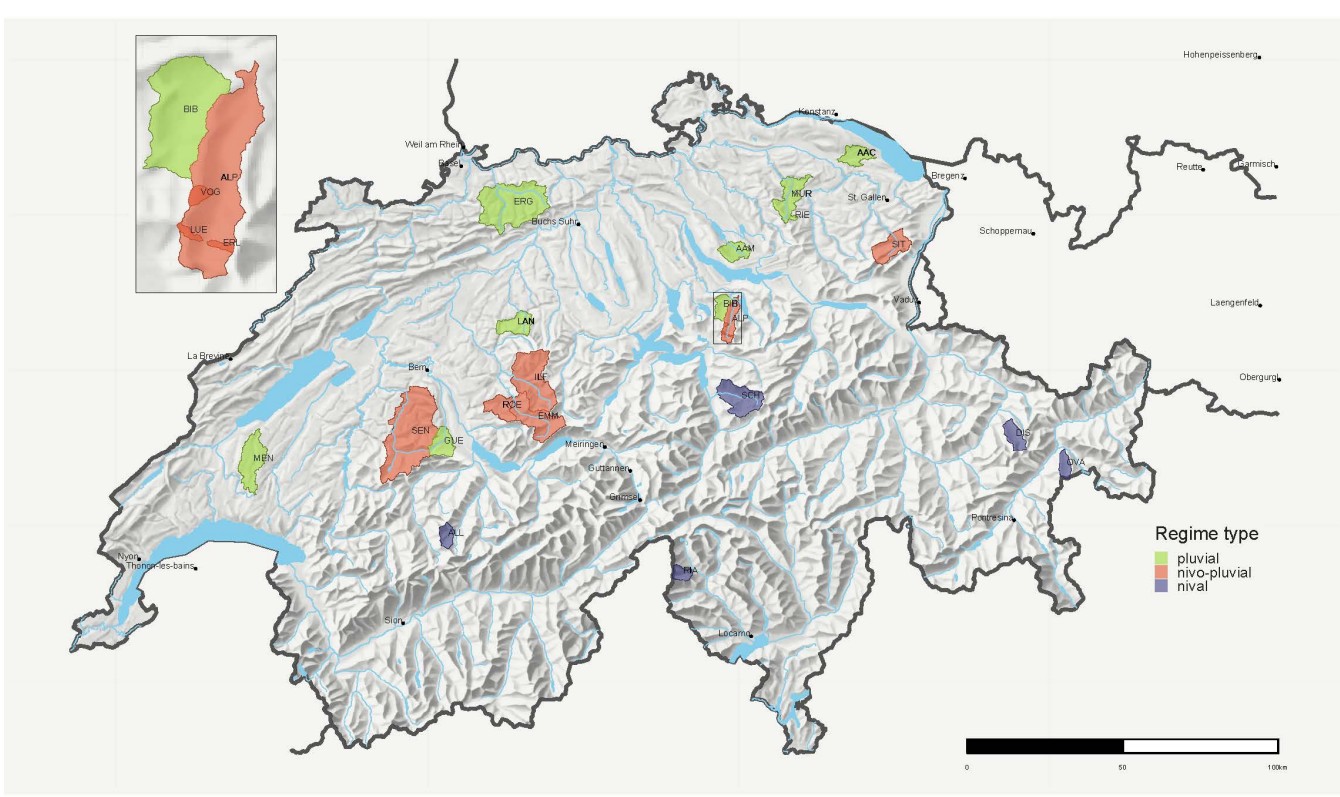





Figure 2






Figure 3



Figure 4

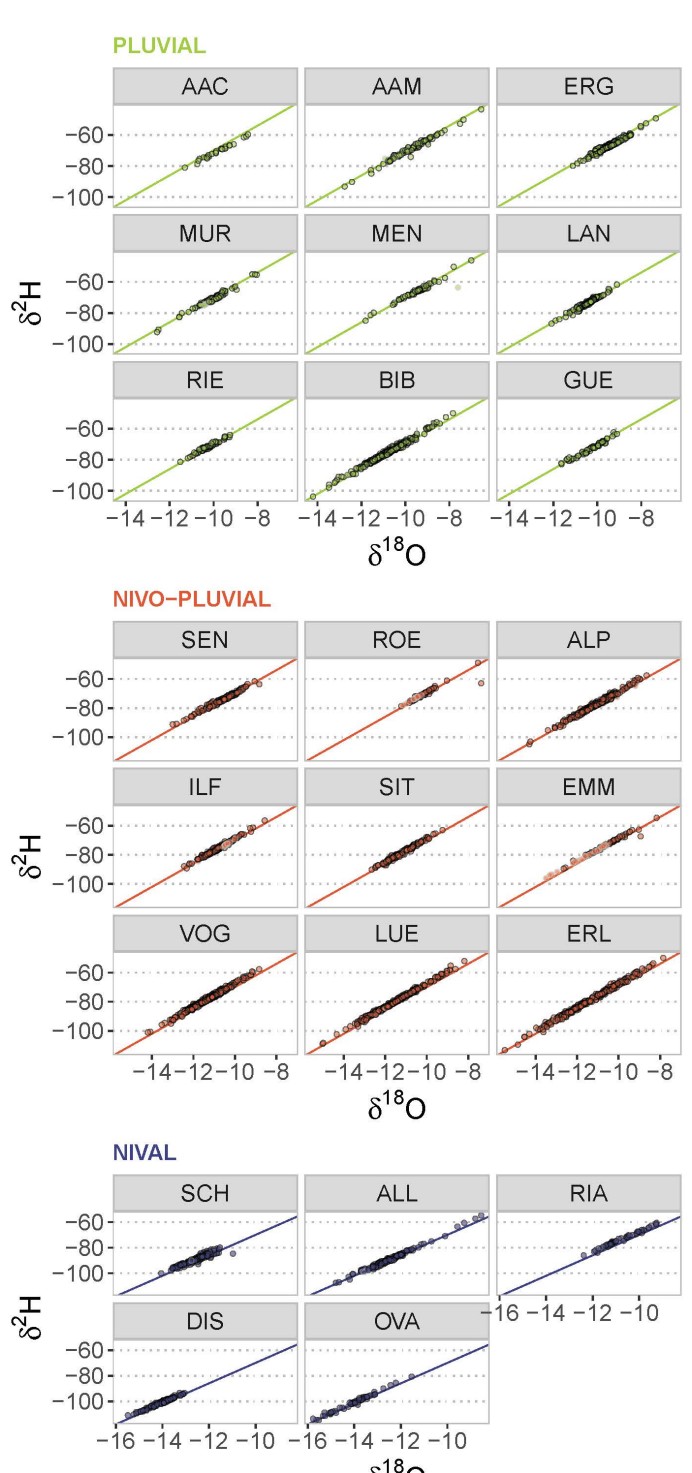



Figure 5

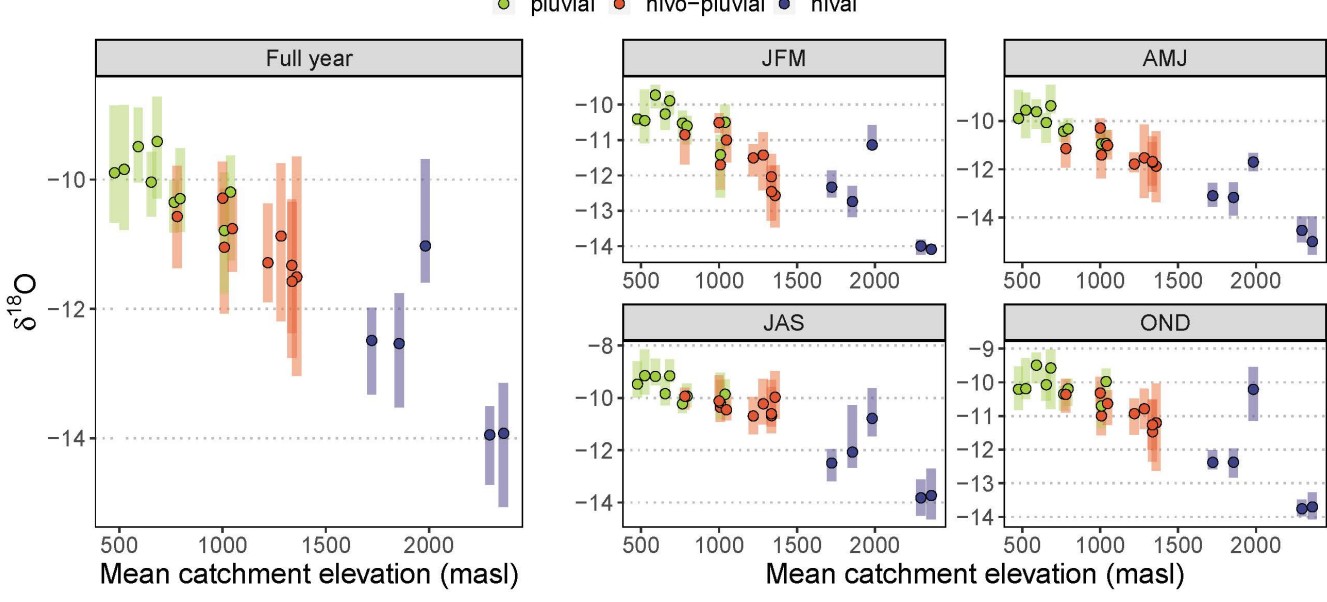



Figure 6

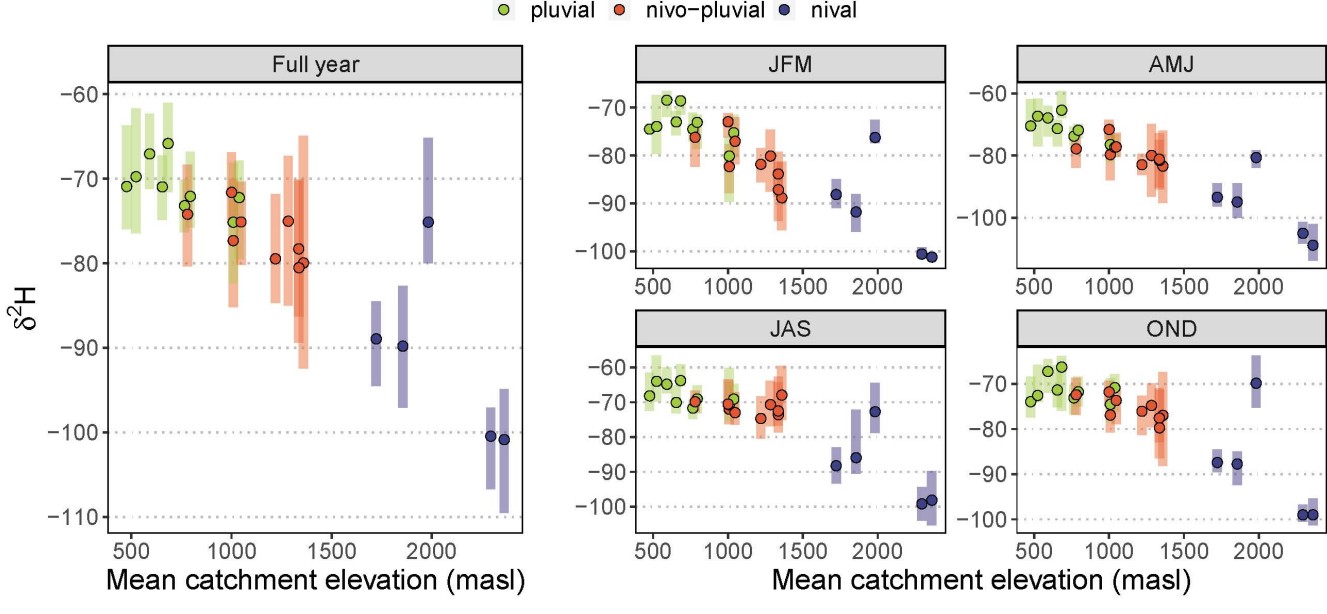


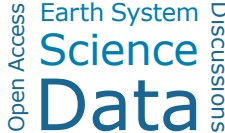

Figure 7

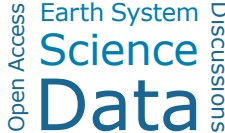

Figure 8



