# Peer review of "The CH-IRP data set: a decade of fortnightly data of $\delta^2$ H and $\delta^{18}$ O in streamflow and precipitation in Switzerland"

_Earth System Science Data, 2020_

## Referee Comment (RC1) · Anonymous Referee #1 · 19 Aug 2020

Major comments: The isotopic compositions of stream water in 23 catchments are indeed interesting to the hydrologists. I also understand the hard work for such collection. However, the main concern for me about this work is that it lacks measured isotopes data of precipitation. Although the authors gave the interpolation values for each catchment, I would love to see measurements of these values. Moreover, the data set did not provide precipitation amount data. As the author mentioned, the data set is useful to estimate mean transit time (MTT) or run catchment hydrological models. However, the precipitation isotopic ratios and amount are the most important inputs for the models. MTT was calculated by modeling the relationship between precipitation as an input and hydrological components as outputs. Consequently, it can not calculate MTT accurately if there is no accurate input information. From my point, the data set did not support the usefulness which mentioned in the introduction. Minor comments: L40 od should be of L159 Average elevation gradients for each month were calculated. Why the average elevation gradients for each month were calculated? I did not understand this sentence.

---

## Author Comment (AC1) · 25 Aug 2020

*We thank the reviewer for the useful comments and include our response to each of the points in italic.*

Major comments: The isotopic compositions of stream water in 23 catchments are indeed interesting to the hydrologists. I also understand the hard work for such collection. *Thank you!*

However, the main concern for me about this work is that it lacks measured isotopes data of precipitation. Although the authors gave the interpolation values for each catchment, I would love to see measurements of these values.

*We also wished we had been able to collect precipitation isotope samples but unfortunately this was not feasible given the available funding. However, we provide the modelled precipitation isotope composition to support other applications.*

Moreover, the data set did not provide precipitation amount data. As the author mentioned, the data set is useful to estimate mean transit time (MTT) or run catchment hydrological models. However, the precipitation isotopic ratios and amount are the most important inputs for the models. MTT was calculated by modeling the relationship between precipitation as an input and hydrological components as outputs. Consequently, it can not calculate MTT accurately if there is no accurate input information. From my point, the data set did not support the usefulness which mentioned in the introduction.

*Unfortunately, for legal reasons we are not allowed to supply the precipitation amounts, as asked for by the reviewer, but these data are freely available from the Swiss Federal Office of Meteorology and Climatology (MeteoSwiss) on request (https://www.meteoswiss.admin.ch/home/services-and-publications/beratung-und-service/datenportal-fuer-lehre-und-forschung.html) and with these the MTTs can be calculated.*

Minor comments: L40 od should be of
*We will change that.*

L159 Average elevation gradients for each month were calculated. Why the average elevation gradients for each month were calculated? I did not understand this sentence.
*We have to clarify that, the gradients are lapse rates that change through the year. The elevation gradients remain the same of course.*
* * *

---

## Referee Comment (RC2) · Anonymous Referee #2 · 26 Aug 2020

The presented data collection is a unique and valuable contribution for the hydrology community. The accompanying text is comprehensible. However, there are essential information missing (rainfall and streamflow data, basin boundaries) to make it a complete data collection that is usable for the community. It should be in the interest of the authors to provide a complete collection, as it will increase the impact of the published data and reduce misinterpretation.

Rainfall data – to make the dataset comprehensive you need to supply the rainfall amounts (e.g. as basin weighted averages), the link provided to Referee 1 is not sufficient at all, it only leads to a german webpage (not available in english). Even

if anyone would be able to go through the registration process it is not at all clear which weather stations are required to obtain data for the specific basins. I guess not each of your basins has a weather station? And the names of corresponding weather stations might not be the names of the basins. I understand that certain institutions will not allow the publication of raw data, but I am convinced that Meteoswiss will allow publication of the data (especially when providing processed basin weighted averages) as part of a scientific publication (see section "how data may be used" https://gate.meteoswiss.ch/idaweb/more.do?language=en)

Streamflow data – Streamflow timeseries are essential for multiple applications, and I would highly recommend to add them to the data collection. In any case, the information that is currently provided is not sufficient. It is not clear where one can obtain the streamflow timeseries. In line 207 you mention Swiss FOEN as a data source (for streamflow and basin boundaries), however the stations ROE, ERL, VOG, LUE, AAM are not existing in the public dataset (https://www.hydrodaten.admin.ch/en/stations-and-data.html). GUE has a different FOEN ID and coordinates. You need to at least provide detailed information where one can get the data for all basins, but as outlined above I think providing streamflow timeseries would be a valuable addition to the present data collection.

Basin boundaries – It would be great if you could provide basin boundaries for all basins, as for most applications these are needed to clip other data sources. E.g. without providing rainfall data, missing basin boundaries are another obstacle for interested users. It's clear that you have them available and I do not see a reason why you should not provide them.

Minor Comments:

Line 80 - replace "stemming from" with sth like "collected in"

Line 141 – In my understanding, the flow percentile that you describe is 0.05, terms are mixed up, I guess you mean the exceedance probability of 0.95 or the percentile of

0.05

Figure 1: increase size of scale and basin IDs

Figure 8: there is a 1 on top of January

––––––––––––––––––––––––––––––

---

## Author Comment (AC2) · 31 Aug 2020

*We thank the reviewer for the valuable comments and include our response below each comment in italic.*

The presented data collection is a unique and valuable contribution for the hydrology community. The accompanying text is comprehensible.

*Thank you for this positive comment.*

However, there are essential information missing (rainfall and streamflow data, basin boundaries) to make it a complete data collection that is usable for the community.

[Figure]

It should be in the interest of the authors to provide a complete collection, as it will increase the impact of the published data and reduce misinterpretation.

*Yes, we agree that precipitation and streamflow data as well as catchment boundaries might be needed to make our data easy to use for other studies, as we also wrote in the manuscript. Where these other data are available from the respective agencies, we will add these data to the revised version of our data paper (see our response to the comment below).*

Rainfall data – to make the dataset comprehensive you need to supply the rainfall amounts (e.g. as basin weighted averages), the link provided to Referee 1 is not sufficient at all, it only leads to a german webpage (not available in english). Even if anyone would be able to go through the registration process it is not at all clear which weather stations are required to obtain data for the specific basins. I guess not each of your basins has a weather station? And the names of corresponding weather stations might not be the names of the basins. I understand that certain institutions will not allow the publication of raw data, but I am convinced that Meteoswiss will allow publication of the data (especially when providing processed basin weighted averages) as part of a scientific publication (see section "how data may be used" https://gate.meteoswiss.ch/idaweb/more.do?language=en)

*Motivated by the request of the reviewer(s), we contacted the respective agencies again and inquired whether we would be allowed to publish the data together with our data set. For the basin weighted average precipitation and we got the OK from MeteoSwiss to do so. We will add these data and the source ("Rhires" gridded data product) to the data set and describe the the basin weighted average precipitation in the manuscript.*

Streamflow data – Streamflow timeseries are essential for multiple applications, and I would highly recommend to add them to the data collection. In any case, the information that is currently provided is not sufficient. It is not clear where one can obtain

the streamflow timeseries. In line 207 you mention Swiss FOEN as a data source (for streamflow and basin boundaries), however the stations ROE, ERL, VOG, LUE, AAM are not existing in the public dataset (https://www.hydrodaten.admin.ch/en/stationsand-data.html). GUE has a different FOEN ID and coordinates. You need to at least provide detailed information where one can get the data for all basins, but as outlined above I think providing streamflow timeseries would be a valuable addition to the present data collection.

*Thank you for carefully checking the stations. We did indeed miss to provide the reference to the Alptal stations (ERL, VOG, LUE) that are maintained by the WSL (Swiss Federal Institute for Forest, Snow and Landscape Research) as well as for the Cantonal stations of ROE and AAM and we will add this information. In Table1, column FOENID there are simply blanks indicating that the stations are not maintained by the Federal Office. We will change that to make clear from where the respective data come. We asked again at the Cantons as well as at the FOEN and the WSL whether we could provide the streamflow time series. Not all gave us the allowance, and we will - for consistency and completeness - within the data set not add the streamflow data to our data set. We will, however, clearly indicate from where the streamflow data of each station can be obtained.*

Basin boundaries – It would be great if you could provide basin boundaries for all basins, as for most applications these are needed to clip other data sources. E.g. without providing rainfall data, missing basin boundaries are another obstacle for interested users. It's clear that you have them available and I do not see a reason why you should not provide them.

*We agree. We have this info as shapefiles and after confirmation of the FOEN we will be allowed to add these shapefiles to our data set (referring to the FOEN as data source).*

Minor Comments:
Line 80 - replace "stemming from" with sth like "collected in"

*We will replace that.*

Line 141 – In my understanding, the flow percentile that you describe is 0.05, terms are mixed up, I guess you mean the exceedance probability of 0.95 or the percentile of 0.05

*Yes, we intended to refer to the exceedance probability. We will change the text accordingly.*

Figure 1: increase size of scale and basin IDs

*OK.*

Figure 8: there is a 1 on top of January

*We will remove that, thanks for spotting the mistake.*

---

## Author Response (AR1)

**Letter to the editor**

Dear David Carlson,

Thank you for your efforts with our manuscript! We changed the manuscript according to the points the reviewers raised and adapted the data on the zenodo repository now including also areal precipitation and temperature as well as the shape files for each catchment (https://zenodo.org/record/4057967). Please, find the point-by-point reply (blue) to the reviewer comments (black) below.

On behalf of all co-authors,

Maria Staudinger
* * *
Reviewer #1

We thank the reviewer for their useful comments and include our response to each of their points here.

Major comments: The isotopic compositions of stream water in 23 catchments are indeed interesting to the hydrologists. I also understand the hard work for such collection.
Thank you!

However, the main concern for me about this work is that it lacks measured isotopes data of precipitation. Although the authors gave the interpolation values for each catchment, I would love to see measurements of these values.
We also wished we had been able to collect precipitation isotope samples but unfortunately this was not feasible given the available funding. However, we provide the modelled precipitation isotope composition to support other applications.

Moreover, the data set did not provide precipitation amount data. As the author mentioned, the data set is useful to estimate mean transit time (MTT) or run catchment hydrological models. However, the precipitation isotopic ratios and amount are the most important inputs for the models. MTT was calculated by modeling the relationship between precipitation as an input and hydrological components as outputs. Consequently, it can not calculate MTT accurately if there is no accurate input information. From my point, the data set did not support the usefulness which mentioned in the introduction.
Motivated by the request of the reviewer(s), we contacted the Swiss federal agency, MeteoSwiss, again and inquired whether we would be allowed to publish the data together with our data set. For the basin weighted average precipitation and we got the OK to do so and now added the areal precipitation to the dataset in the data repository.

Minor comments:
L40 od should be of
We changed that.

L159 Average elevation gradients for each month were calculated. Why the average elevation gradients for each month were calculated? I did not understand this sentence.
We rephrased this part in the manuscript. The gradients are lapse rates that change through the year. The elevation gradients remain the same of course.

Reviewer #2

The presented data collection is a unique and valuable contribution for the hydrology community.
The accompanying text is comprehensible.
Thank you for this positive comment.

However, there are essential information missing (rainfall and streamflow data, basin boundaries) to
make it a complete data collection that is usable for the community. It should be in the interest of
the authors to provide a complete collection, as it will increase the impact of the published data and
reduce misinterpretation.
Yes, we agree that precipitation and streamflow data as well as catchment boundaries might be
needed to make our data easy to use for other studies, as we also wrote in the manuscript. Where
these other data are available from the respective agencies, we added these data to the revised
version of our data paper (see our response to the comment below).

Rainfall data – to make the dataset comprehensive you need to supply the rainfall amounts (e.g. as
basin weighted averages), the link provided to Referee 1 is not sufficient at all, it only leads to a
german webpage (not available in english). Even if anyone would be able to go through the
registration process it is not at all clear which weather stations are required to obtain data for the
specific basins. I guess not each of your basins has a weather station? And the names of
corresponding weather stations might not be the names of the basins. I understand that certain
institutions will not allow the publication of raw data, but I am convinced that Meteoswiss will allow
publication of the data (especially when providing processed basin weighted averages) as part of a
scientific publication (see section "how data may be used"
https://gate.meteoswiss.ch/idaweb/more.do?language=en)
Motivated by the request of the reviewer(s), we contacted the respective agencies again and
inquired whether we would be allowed to publish the data together with our data set. For the basin
weighted average precipitation and we got the OK from MeteoSwiss to do so. We added these data
and the source ("Rhires" gridded data product) to the data set and describe the basin weighted
average precipitation in the manuscript.

Streamflow data – Streamflow timeseries are essential for multiple applications, and I would highly
recommend to add them to the data collection. In any case, the information that is currently
provided is not sufficient. It is not clear where one can obtain the streamflow timeseries. In line 207
you mention Swiss FOEN as a data source (for streamflow and basin boundaries), however the
stations ROE, ERL, VOG, LUE, AAM are not existing in the public dataset
(https://www.hydrodaten.admin.ch/en/stationsand-data.html). GUE has a different FOEN ID and
coordinates. You need to at least provide detailed information where one can get the data for all
basins, but as outlined above I think providing streamflow timeseries would be a valuable addition to
the present data collection.
Thank you for carefully checking the stations. We did indeed miss to provide the reference to the
Alptal stations (ERL, VOG, LUE) that are maintained by the WSL (Swiss Federal Institute for Forest,
Snow and Landscape Research) as well as for the Cantonal stations of ROE and AAM and we will add
this information. In Table1, column FOENID there are simply blanks indicating that the stations are
not maintained by the Federal Office. We will change that to make clear from where the respective
data come. We asked again at the Cantons as well as at the FOEN and the WSL whether we could
provide the streamflow time series. Since not all gave us the allowance, we decided for consistency
and completeness within the data set to not add the streamflow data to our data set. However, we
indicated now more clearly from where the streamflow data of each station can be obtained.

Basin boundaries – It would be great if you could provide basin boundaries for all basins, as for most applications these are needed to clip other data sources. E.g. without providing rainfall data, missing basin boundaries are another obstacle for interested users. It's clear that you have them available and I do not see a reason why you should not provide them.
We agree. We added these shapefiles to our data set (referring to the FOEN as data source).

Minor Comments:
Line 80 - replace "stemming from" with sth like "collected in"
We replaced that.

Line 141 – In my understanding, the flow percentile that you describe is 0.05, terms are mixed up, I guess you mean the exceedance probability of 0.95 or the percentile of 0.05
Yes, we intended to refer to the exceedance probability. We changed the text accordingly.

Figure 1: increase size of scale and basin IDs
We increased the size as recommended.

Figure 8: there is a 1 on top of January
We removed that, thanks for spotting the mistake.

[revised manuscript text omitted]

Figure 8 Monthly maps of sea level corrected $\delta$18O in precipitation.

Figure 1

[Figure]

[Figure]

Figure 2

[Figure]

380   Figure 3

[Figure]

Figure 4

[Figure]

[Figure]

Figure 6

[Figure]

Figure 7

[Figure]

Figure 8

[Figure]

sea level corrected precipitation δ¹⁸O [‰]

○ measurement site    × validation site    ◇ target site

sea level corrected precipitation $\delta^{18}$O [‰]

○ measurement site   ✕ validation site   ◇ target site

395

---

## Author Response (AR2)

**Letter to the editor**

Dear David Carlson,

We can understand that it might be less comfortable to not receive the streamflow data directly together with the isotopes and precipitation. However, as we already stated in our last reply, if we added the data for some stations and not for others that would be inconsistent and further that would not include data homogenization updates that the agencies perform from time to time. Hence, we think the effort to order the streamflow data for interested readers should be no problem with the links provided.

All staff of the cantonal and federal agencies is capable of understanding orders in English, and we think that a template letter to order data is not necessary.

We added links / email contacts to order streamflow data. We also revised carefully the names and IDs of the catchments in the manuscript, and together with the added links they provide enough information to be able to order the corresponding streamflow data for every catchment in our data set.

Please, find also a point by point response to the reviewer comments below.

On behalf of all co-authors,

Maria Staudinger

Reviewer comments

Thank you for implementing the changes and providing additional data.

It is still somewhat bothersome that streamflow data is not directly available (apart from WSL data). It will be really painful for non-german speakers, especially coming from non-European countries, to obtain data from different Swiss authorities. However, I understand that this is beyond your control to some extent. In your reply regarding the streamflow data availability you state that "not all gave us allowance". Please consider the following:

- add streamflow data for all catchments where you have the permission and clearly state for which authorities it was not granted

We can understand that it might be less comfortable to not receive the streamflow data directly together with the isotopes and precipitation. However, as we already stated in our last reply, if we added the data for some stations and not for others that would be inconsistent and further that would not include data homogenization updates that the agencies perform from time to time.

- for these authorities, provide detailed information where to request data including links to the respective pages to facilitate access

We added a link /email address to each agency/office to facilitate access (L222-231).

- add an example letter in German and English in the supplement, addressed to the right authorities including the station names, IDs and all information that is required to obtain the data

We know that all staff of the cantonal and federal agencies is capable of understanding English, and we think that a template letter to order data is not necessary.

I understand that this might involve some further changes and additional work, but consider this a huge favor to our community, facilitating open access to a broad international audience.

Please increase the size of the "regime type" legend in Figure 1.

We changed that.

[revised manuscript text omitted]

Figure 8 Monthly maps of sea level corrected $\delta 18O$ in precipitation.

Figure 1

[Figure]

[Figure]

375

Figure 2

[Figure]

380   Figure 3

[Figure]

Figure 4

[Figure]

385   Figure 5

[Figure]

Figure 6

[Figure]

Figure 7

[Figure]

Density

Sampling Q index

Figure 8

[Figure]

sea level corrected precipitation $\delta^{18}$O [‰]

○ measurement site    × validation site    ◇ target site

395